# Effect of Injection Overmolding Parameters on the Interface Bonding Strength of Hybrid Thermoset–Thermoplastic Composites

**DOI:** 10.3390/polym15132879

**Published:** 2023-06-29

**Authors:** Tianyu Miao, Wenhao Wang, Zhanyu Zhai, Yudong Ding

**Affiliations:** 1College of Mechanical and Electrical Engineering, Central South University, Changsha 410083, China; 2State Key Laboratory of Precision Manufacturing for Extreme Service Performance, Central South University, Changsha 410083, China

**Keywords:** hybrid thermoset–thermoplastic composite, injection overmolding, molecular dynamic simulation

## Abstract

In this study, the thermoset–thermoplastic structure was produced through a co-curing technique together with an injection overmolding technique. Continuous fiber reinforced thermoset composite (TSC) was selected as thermoset material, while polyamide 6 (PA 6) was chosen as thermoplastic material. The influence of injection temperature, preheating temperature and injection speed on the interfacial bonding strength of hybrid thermoset–thermoplastic composites was investigated. The results show that increasing injection temperature and preheating temperature have significant effects on the increase in bonding strength, while injection speed has little effect on it. In addition, the bonding strength of the co-cured interface is enhanced after the injection overmolding process, which is further studied through molecular dynamic (MD) simulation. The molecular dynamic simulation result shows that the high temperature and pressure during the injection process only have a weak effect on enhancing the bonding strength of the co-cured interface, while the chemical reaction at the co-cured interface is the main reason for the enhancement. Furthermore, the more chemical reactions occur at the interface, the stronger the interface will be.

## 1. Introduction

Lightweight is a requirement for energy conservation and emission reduction [1,2]. The use of lightweight materials is an effective method to achieve lightweight in the automotive and aerospace areas [3,4]. As a lightweight material with bright development prospects, continuous fiber reinforced thermoset composite (TSC) has high heat and corrosion resistance and excellent mechanical properties [5,6,7,8]. However, on the other hand, it shows difficulties in the production of parts. Specifically, TSC is mainly molded by hot press and autoclave, which undoubtedly increases the difficulty of forming parts with complex geometry. Based on this, a multi-material to form TSC together with thermoplastic can combine the high performance of TSC with the formability of complex geometries of thermoplastic.

At present, the most common assembly methods between two different materials are mechanical fastening, adhesive bonding [9], welding [10] (self-resistance welding and ultrasonic welding) and injection overmolding. Among the above assembly methods, except for injection overmolding, other methods firstly require separate molding of TSC and thermoplastic structure, and then realize the assembly of the two structures through a new process. The additional assembly process brings about the demand for additional labor and increase in cycle time and cost. In contrast, placing TSC into the injection mold and directly bonding the TSC to the thermoplastic material through injection overmolding shows greater potential in the reduction of cycle time and labor costs [11].

The interfacial bonding strength of composite structures is an important index to evaluate the performance of thermoset–thermoplastic structures, thus many researches focus on improving the interfacial bonding strength through various methods. Kazan et al. [12,13,14] obtained prepregs with different degrees of curing by changing the preheating time, and then overmolded thermoplastic polypropylene (PP) on the CF/epoxy prepreg with different curing degrees through an injection process. It is found that the maximum flexural strength of the specimens occurs at the maximum value of preheating time and injection temperature. Karakaya et al. [15,16] found that, the introduction of peel-ply and preheating of the TSC can contribute towards forming a good interface between the TSC and polyamide 6 (PA 6) by overmolding. Furthermore, the application of the hot melt adhesive between the TSC and PA 6 is also conducive to the improvement of bonding strength. Ding et al. [17] proposed that surface pretreatments (plasma treatment, surface silanization and CO_2_ laser ablation) can effectively improve the bonding strength of overmolded thermoset–thermoplastic structures, especially laser treatment. Based on these studies, a method to form a TSC-PA 6 structure by co-curing together with an overmolding process was proposed in our previous research [18]. The bonding strength of the co-cured interface was increased by applying plasma treatment, increasing the roughness of thermoplastic film and applying thermoplastic film with lower melt temperature. Finally, a TSC-PA 6 structure with high bonding strength was formed by means of the overmolding process. This study proves that the co-curing process together with overmolding process has good application prospects in forming TSC-PA 6 structures.

Our previous research [18] mainly focuses on improving the interfacial bonding strength of the co-cured interface, while the final bonding strength of the TSC-PA 6 structure is also affected by the injection overmolding process parameters. Therefore, one of the aims of this research is to study the influence of injection overmolding parameters on the interfacial bonding strength of overmolded hybrid TSC-PA6 structures. For overmolded thermoplastic-thermoplastic interfaces, many researches have investigated the bonding mechanism through reptation theory [19,20,21,22]. However, for thermoset–thermoplastic interfaces, the bonding mechanism changes due to the inability of thermoset materials to remelt at high temperature. Therefore, another goal of this research is to explore and explain the mechanism of the enhanced bonding strength of co-cured interfaces.

Molecular dynamic simulations have been widely used in explaining the experimental phenomena [23] and have become an important means of studying interfacial properties. Molecular dynamic simulations have been applied to study the interface performance between two different materials, such as the metal–metal interface [24], polymer–metal interface [25], polymer–fiber interface [26] and polymer–polymer interface. Among them, the research on the polymer–polymer interface cover the thermoplastic–thermoplastic interface and thermoset–thermoplastic interface. Jiang et al. [27] studied the effect of injection temperature and pressure on the properties of the PA 66-PP interface formed by injection overmolding by molecular dynamics simulations, and the formation and failure of the interface were characterized. Laurien et al. [28] established an interface model between poly(vinylidene difluoride) (PVDF) and multicomponent epoxy resin. The forming mechanism of the interface during the co-curing process was revealed, and the failure of the interface was studied through tensile simulation. In summary, molecular dynamic simulation has become an important tool for studying interfacial properties, thus it is applied as the main means to study the interface bonding mechanism in this study.

This study investigated the effect of injection overmolding parameters on the interfacial bonding strength of TSC-PA 6 structures through experimental methods, and the mechanism of bonding strength enhancement of co-cured interfaces was studied through molecular dynamics simulations. Specifically, TSC-PA 6 structures were firstly formed through a co-curing process followed by injection overmolding process. After that, the micromorphology of the bonding area of the structure was observed through ultra-depth-of-field microscopy, while the lap shear strength of the structures was obtained by tensile testing to characterize the bonding quality of the interface. Moreover, the experimental phenomenon of the increase in the bonding strength of co-cured interface after injection overmolding process was studied through molecular dynamics simulations.

## 2. Experimental Setup

### 2.1. Materials

The co-cured structure as an insert for injection overmolding is composed of two parts: TSC and a layer of film. Specifically, the TSC was prepared from unidirectional carbon fiber prepreg (USN10000, Guangwei Group Co., Ltd., Weihai, China). The resin content was 40% and the thickness was 10 mm for one layer of prepreg. Furthermore, the resin was diglycidyl ether of bisphenol-A (DGEBA), and the curing agent was dicyandiamide (DICY). The PA 6 film (Ultramid B3S, BASF Co., Ltd., Ludwigshafen, Germany) was selected as the layer. PA 6 granules (7331J NC010, Dupont, Wilmington, NC, USA) were selected as injection material for overmolding. Epoxy resin (EPIKOTE 828, Hexion, Columbus, USA) without other chemical components was used for investigating interfacial reaction between the epoxy and PA 6.

### 2.2. Sample Preparation

The hybrid TSC-PA 6 structure was prepared in two steps, namely, the co-curing process and overmolding process, as shown in Figure 1.

#### 2.2.1. Preparation of Co-Cured Structure

In our previous study [18], the PA 6 films were finally selected as the thermoplastic layer in the co-cured structure by comparing them with other thermoplastic candidates. On this basis, the co-cured structure is designed as seen in Figure 1, which consists of a layer of PA 6 film and 12 layers of prepreg. The stack was co-cured using a hot-press with the process parameters recommended by manufacturer (130 °C and 0.6 MPa for 90 min). The size of the co-cured structure was 130 mm × 130 mm, and the thickness was 1.5 ± 0.1 mm. The co-cured structure was then cut to a size of 100 mm × 10 mm for the overmolding process.

#### 2.2.2. Overmolding of PA 6

After drying the PA 6 granules at 80 °C for 6 h, the co-cured structure was put into injection mold as an insert and molten PA 6 was injected on its surface using an injection machine (ZE1200 III/300, Ningbo Haitian Plastic Machinery Group Co., Ltd. Ningbo, China). During the overmolding process, the mold temperature was controlled at 80 °C by the mold temperature machine (AEOT-20-24, Aode Machinery Co., Ltd. Shanghai, China). The surface of the co-cured structure was heated by an infrared heating device until it reached the required temperature. Since both the thermoplastic film and the injection part are PA 6, the bonding performance of the interface can be investigated based on the reptation theory [20]. According to the reptation theory, the degree of healing is related to the interfacial bonding strength, and temperature related parameters are the main factors affecting the degree of healing [21]. Therefore, this research mainly studies the effect of injection temperature, preheating temperature and injection speed (may affect the consumption of temperature during the injection process) on the interfacial bonding strength. Six different groups of parameters were set as shown in Table 1. The final overmolded structure is shown in Figure 2.

### 2.3. Characterization

#### 2.3.1. Morphology Characterization

The cross section of the bonding area of the overmolded sample was observed using ultra-depth-of-field microscopy (VHX-5000, Keyence Co., Ltd., Osaka, Japan). The bonding area was firstly cut from the overmolded structure, then it was mosaiced and polished with a metallographic grinding machine. After polishing the surface, the cross section of the sample was observed through a microscope.

#### 2.3.2. Lap Shear Tests

The bonding strength of the overmolded structure was characterized by lap shear test. The lap shear strength was measured using an electronic universal testing machine (CMT 4204, MTS Co., Ltd., Eden Prairie, MN, USA) with a load cell of 30 kN. In this testing process, the overmolded structure was fixed on the fixture as shown in Appendix A, and the displacement loading was carried out at a rate of 1 mm/min. The lap shear strength was calculated using Equation (1).
(1)$$\tau =\frac{F}{d\times l}$$
where F represents the max loading force, and d and l represent the width and length of the bonding area, respectively.

#### 2.3.3. Fourier Transform Infrared (FTIR) Spectroscopy

The chemical reaction between epoxy resin and PA 6 was investigated using Fourier transform infrared spectroscopy (Nicolet iS50, Thermo Fisher Scientific Co., Ltd., Waltham, MA, USA). First, the epoxy resin was dissolved in an acetone solution, then the PA 6 film was immersed in the solution for 1 min to form a layer of epoxy resin on it. The sample was placed into an oven and heated at 130 °C and 200 °C for 1 min, respectively (to investigate whether interface reactions will occur under co-curing and injection process parameters, respectively). After that, the surface of the sample was washed with acetone to clean off the unreacted epoxy resin. Finally, the surface of the modified PA 6 film was analyzed using Fourier transform infrared spectroscopy and the wavenumber range of 400–4000 cm^−1^ was recorded.

### 2.4. Simulation Procedure

Molecular dynamics simulation was applied to investigate the mechanism of enhancement of the co-cured interface after the injection overmolding process.

#### 2.4.1. Model Construction

The simulation in this study will mainly focus on the TSC–PA6 interface to investigate the bonding mechanism of the co-cured interface. The effects of high temperature and pressure during the injection process, as well as interfacial chemical reactions at high temperatures, on the bonding strength of the thermoset–thermoplastic interface were investigated. The fiber bundle is wrapped with epoxy resin in the TSC structure, which actually contacts the PA 6 at the interface. Thus, the molecular model of the interface between TSC and PA 6 was built simply as the interface between crosslinked epoxy resin and PA 6, as shown in Figure 3.

PA 6 was chosen as the thermoplastic material according to the experiment. The PA 6 molecular model was composed of 24 chains with 16 repeating units in each chain, and the density of the model is 1.13 g/cm^3^. The size of the PA 6 layer was 4.5 × 4.5 × 3 nm in x, y and z directions, respectively.

The crosslinked epoxy resin model was built as a mixture consisted of diglycidyl ether of bisphenol-A (DGEBA) as epoxy resin and dicyandiamide (DICY) as curing agent, and the molecular number ratio of DGEBA to DICY was 1:1. The size of the epoxy layer was 4.5 × 4.5 × 3 nm in x, y and z directions and the density was 1.2 g/cm^3^. It indicates that the reactive groups in crosslinked DGEBA are mainly epoxy and hydroxyl groups. Therefore, when establishing a crosslinked epoxy resin model, the consumption of epoxy groups and the generation of hydroxyl groups during crosslinking are mainly considered, and the crosslink reaction is simplified as shown in reaction Formula (4) [29]. The subsequent chemical reaction between PA 6 and crosslinked DGEBA are mainly as shown in reaction Formulas (2) and (3) [30].
(2)
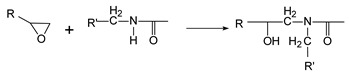

(3)
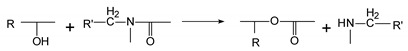

(4)
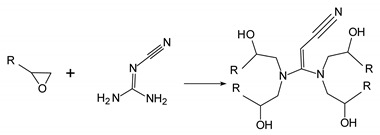


The crosslink reaction process is shown in Figure 4. First, the established DGEBA and DICY mixture and the initial reaction radius were input, then the geometric optimization and dynamic relaxation (under a NVT ensemble with 400 K) was applied to the model. After that, it was determined whether there were reactive atomic pairs in the reaction radius. If not, the reaction radius would increase. If yes, the crosslink reaction as shown in (4) was realized. The above reaction steps repeated until the reaction radius reached the set maximum reaction radius. Finally, geometric optimization and dynamic relaxation were performed on the model that completed the crosslink reaction. Furthermore, the degree of curing is an important index for evaluating the crosslink reaction of epoxy resin. The degree of curing is defined as the percentage of the released reaction enthalpy over the total released reaction enthalpy, and the released reaction enthalpy is related to the number of reaction atoms. Thus, for the crosslinked epoxy resin model, the degree of curing is defined as the ratio of the number of reactive atoms to the total number of reactive atoms. In this study, a maximum reaction radius of 12 A was set and a crosslinked epoxy resin model with a degree of curing of 93% was ultimately obtained.

#### 2.4.2. Simulation Procedure

The simulation consists of two parts, the bonding between crosslinked epoxy resin and PA 6, and uniaxial tensile simulation of the bonded interface. The consistent valence force field (CVFF), which has been widely used in the study of polymers [31,32], was applied to calculate the intermolecular and non-bonded interactions between atoms. In this work, the forming of the interface was mainly realized through simulation in Materials Studio, while reactions were implemented using Perl scripts. Furthermore, the LAMMPS molecular dynamics simulator was used for interface separation.

##### Bonding between Crosslinked Epoxy Resin and PA 6

To study the mechanism of the increase in the strength of the co-cured interface after injection overmolding, four groups of co-cured interfaces were set up as shown in Table 2, namely, C, O-unreacted, O-reacted (53.6%) and O-reacted (100%). The high temperature and pressure during the injection process, as well as the chemical reactions occurring at the interface, were considered as the possible cause of the enhancement of the co-cured interface. Of these four interface models, the C and O-unreacted were used for studying the effects of high temperature and high pressure, while O-unreacted, O-unreacted (53.6%) and O-reacted (100%) were used for studying the influence of interfacial chemical reaction.

The co-curing process simulation was performed under an NPT (i.e., constant number of atoms, pressure and temperature) ensemble with the pressure of 0.6 MPa for all groups. Firstly, the boxes were equilibrated at the temperature of 300 K for 100 ps. Then it took 150 ps to heat up the boxes from 300 to 400 K. Next, the boxes were equilibrated at 400 K for 150 ps and cooled down from 400 to 300 K for 150 ps. Finally, the boxes were equilibrated at 300 K for 10 ps.

The chemical reaction at the interface occurs accompanied by high temperature and pressure. Here, the process was simplified as a chemical reaction followed by applying temperature and pressure to the interface. In order to simulate the chemical reactions that may occur at the interface, the script was used to realize the chemical reaction at the interface for groups O-reacted (53.6%) and O-reacted (100%), and the specific chemical reactions are shown in chemical formulas (2) and (3). For group O-reacted (53.6%), a smaller reaction radius was applied, while for group O-reacted (100%) a larger reaction radius was applied. The degree of reaction was characterized by the number of reaction atoms, as shown in Table 1. Reaction (1) requires epoxy groups as reactants, which are less present in the crosslinked epoxy resin. Thus, it is extremely difficult for reaction (1) to occur at the interface, and reaction (3) becomes the main reaction at the interface. The number of times that reaction (3) occurs at the interface of O-reacted (53.6%) and O-reacted (100%) groups are 15 and 28, while only 1 reaction (1) occurs at the interface of both groups.

Temperature and pressure were further applied to O-reacted, O-reacted (53.6%) and O-reacted (100%) groups to simulate the injection process under the NPT ensemble. Specifically, the pressure was kept at 20 MPa. The boxes were firstly heated up from 300 to 500 K for 150 ps, then the boxes were relaxed at 500 K for 150 ps, finally the temperature was reduced to 300 K after 150 ps, and the structures were relaxed at 300 K for 10 ps.

##### Interface Separation

After the bonding between crosslinked epoxy resin and PA 6, uniaxial tensile simulation was applied to study the mechanical properties and the failure mechanism of the interface. The top layer of PA 6 was set as the rigid body to apply displacement in this area. Uniaxial tensility was achieved by applying a velocity of 0.1 nm/ps along the Z direction to the top rigid body under a constant NVT ensemble at 300 K. Nosé–Hoover thermostat was applied to control the temperature of the interface. Newton’s equations of motion were integrated via the Verlet velocity algorithm, and the free PA 6 body was calculated via the virial theorem.

## 3. Results and Discussion

### 3.1. Microstructures of Injection Overmolding Interface

Figure 5a–f illustrates the cross-section optical micrographs of the interface between the PA 6 film and injection-overmolded PA 6. In all the figures, the top part is the PA 6 film, and the bottom part is the injection-overmolded PA 6. Figure 5a shows the overmolded interface of the L sample (with a lower injection temperature). As seen, there is no effective interface. The large gap between the PA 6 film and the injection-overmolded PA 6 also indicates poor interfacial performance. The overmolded interfaces of M1, M2 and M3 samples are shown in Figure 5b–d. As the injection temperature increase, there are no obvious large gaps between the PA 6 film and the injection-overmolded PA 6. However, there are still small cracks at the interface, as shown in the selected box areas in Figure 5b–d. Such small cracks disappear by further increasing the injection temperature or applying higher preheating temperature, as shown in Figure 5e–f (H sample). Thus, increasing the injection temperature or preheating temperature is beneficial for forming a better interface, while the influence of injection speed is relatively weak.

### 3.2. Single Lap Shear Strength of Hybrid TSC-PA 6 Structure

Figure 6 presents the lap shear strength of overmolded TSC-PA 6 structure fabricated with different process parameters. It is visible that the lap shear strength varies greatly with the process parameter. In particular, the lap shear strengths of H, M1, M2, M3 and M4 samples are 7.1 ± 1.8 MPa, 6.1 ± 0.9 MPa, 6.3 ± 1.1 MPa, 24.4 ± 4.2 MPa and 14.9 ± 2.0 MPa, respectively. It should be noted that the lap shear strength of the L sample is too low to obtain due to the low interface bonding strength, and the result can also correspond to the microscopic morphology of the L sample’s overmolded interface. The intention of forming a TSC–PA 6 structure by co-curing and overmolding process is to achieve high bonding strength by the remelting property of PA 6 film under high temperature, and the lap shear test results can well confirm this conjecture. Specifically, the H sample with higher injection temperature and M4 sample with higher preheating temperature show higher lap shear strength than other samples. This result can be explained by the reptation theory. According to the reptation theory regarding semi-crystalline polymers, the interface temperature is a key factor affecting the bonding performance [21]. The injection temperature and preheating temperature can directly affect the interface temperature, therefore they become important factors affecting the interfacial bonding strength. Increasing the injection speed can reduce the temperature consumption during the injection process. However, the lap shear test results show that the impact of injection speed on the lap shear strength in this experiment is very weak. This may be due to the small volume of injection area, resulting in a small temperature consumption during the injection process under different injection speeds. In addition, in our previous study [18], the maximum lap shear strength of the co-cured interface between PA 6 film and TSC was 10.7 ± 0.3 MPa, which is far less than 24.4 ± 4.2 MPa of the M4 sample. The result indicates that the bonding strength of the co-cured interface may be enhanced after injection overmolding. The detailed mechanism will be discussed in Section 3.4.2.

Figure 7 shows the fracture modes of lap shear test samples after testing. It can be seen from the figure that, except for the H sample and M4 sample, the fracture of other samples all occur at the overmolded interface, and the PA 6 film is not damaged. As for the H sample and M4 sample, the fracture mode occurs at both the co-cured interface (between TSC and PA 6 film) and overmolded interface (between PA 6 film and injection PA 6). Specifically, the bottom of the interface area of the M4 sample fails from the overmolded interface, while the PA 6 film on the top area of the interface is damaged and torn off from the TSC. A similar phenomenon also occurs on sample H, but compared with the M4 sample, the co-cured PA 6 film of the H sample is not damaged. The fracture mode of both the H and M4 samples shows that the top area of the overmolded interface of the M4 sample has a higher bonding strength than other areas, and the bonding strength of the co-cured interface is greater than that of the overmolded interface.

### 3.3. Analysis of FTIR Spectroscopy Results

The FTIR spectra are illustrated in Figure 8. It can be seen from the figure that, compared with the sample treated at 130 °C, a new peak appears at 1720 cm^−1^ for the sample treated at 200 °C. According to previous studies [30,33], the peak at this position indicates the presence of an ester group, which is the product of reaction (3). As the hydroxyl group in the reactant of reaction (3) needs to be obtained by reaction (1), the existence of reaction (1) can also be confirmed. Furthermore, the peak at 916 cm^−1^ indicates the presence of epoxy rings [31], and the peak may represent the epoxy resin being connected to the surface of the PA 6 film by chemical reactions (there are epoxy groups on both sides of DGEBA, and the reaction may only occur at only one side). Furthermore, the peak at 820 cm^−1^ representing the p-disubstituted benzene ring can also confirm this result.

### 3.4. Bonding Mechanism of TSC-PA 6 Interface

#### 3.4.1. Interfacial Bonding Energy between PA 6 and Epoxy Resin

Interface energy is an index for evaluating interfacial bonding strength. The interfacial bonding energy is calculated by the difference between the potential energy of the entire system and its components (PA 6 and epoxy resin), as shown in Equation (5).
(5)$$E_{interface}=E_{system}-\left(E_{PA}+E_{Epoxy}\right)$$
where $E_{interface}$ is the interfacial bonding energy, $E_{system}$ is the energies of the whole system, while $E_{PA}$ and $E_{Epoxy}$ are the energies of the PA 6 and crosslinked epoxy resin layers, respectively.

Figure 9 gives the interfacial bonding energy of co-cured interface models. Specifically, the interfacial bonding energy of C, O-unreacted, O-reacted (53.6%) and O-reacted (100%) interface models are −583 ± 58, −611 ± 58, −836 ± 74 and −1336 ± 56 kcal/mol, respectively. It can be obtained by comparing the interfacial bonding energy of C and O-unreacted that the high temperature and pressure of the injection process will not improve the interfacial bonding energy. This result also indicates that the significant increase in bonding strength of the co-cured interface after injection process is not due to the higher temperature and pressure. It can be concluded by comparing the interfacial bonding energy of the O-reacted (53.6%) and O-unreacted that, interfacial chemical reactions can improve interfacial bonding energy. Moreover, the interfacial bonding energy can be further improved by increasing the number of chemical reactions. Thus, the improved bonding strength of the co-cured interface after injection process is more likely to result from the occurrence of interfacial chemical reaction.

#### 3.4.2. Uniaxial Tensile Deformation Process

##### Tensile Stress–Strain Curve

The tensile stress vs. strain curves for different co-cured interface models are presented in Figure 10. As can be seen, the tensile stress increases almost linearly with engineering strain in the initial stage. For C and O-unreacted models, the tensile stress decreases slowly to a value close to zero after reaching the maximum tensile stress. While for O-reacted (53.6%) and O-reacted (100%) models, the tensile stress decreases slowly and remains at a large value. The main reason for this phenomenon is the introduction of chemical bonds at the interface, which prevents the complete separation of epoxy resin and PA 6. The tensile stresses of C and O-unreacted models are 241.2 MPa and 250.7 MPa, respectively. It indicates that the influence of temperature and pressure on the interfacial bonding strength is relatively small. Furthermore, the tensile strength of the O-reacted (53.6%) model is 291.9 MPa, which is significantly improved compared to the tensile stress of the O-unreacted model. Moreover, as the number of interfacial chemical reactions increases, the tensile strength of the O-reacted (100%) model further increases to 317.6 MPa. It indicates that the occurrence of interfacial reactions can significantly increase the interfacial bonding strength. The result proves that chemical reactions are the main reason for the enhancement of the co-cured interface.

##### Failure Mode

Figure 11 gives the snapshots of the final deformation of the co-cured interface models. As can be seen, there are two types of failure mode. For C and O-unreacted models, adhesive failure is the main failure mode. Specifically, the failure occurs only at the interface between PA 6 and crosslinked epoxy resin. It indicates that the bonding strength of interface is lower than the strength of the polymer itself. In addition, the failure modes of the C and O-unreacted interfaces are the same, which may be due to the cross-linked epoxy resin structure not being conducive to the diffusion of PA 6 at the interface during the interface bonding process. For O-reacted (53.6%) and O-reacted (100%) models, the failure mode changes from adhesive failure to a mix failure mode, which consists of both adhesive and cohesive failure. Specifically, the failure occurs both at the interface and within the polymer itself, which means a higher interfacial bonding strength. The result indicates that the interfacial chemical reaction has a significant impact on the failure mode.

## 4. Conclusions

The influence of injection overmolding parameters on the interfacial bonding performance of TSC–PA 6 structure was investigated through experimental methods, and the mechanism of bonding between PA 6 and crosslinked epoxy resin was investigated through molecular dynamics simulations. The main conclusions are as follows:
Increasing the injection temperature and preheating temperature are beneficial for reducing cracks at the overmolded interface and forming a better interface, while the influence of injection speed is relatively small. Lap shear test results indicate that, the increase in injection temperature and preheating temperature has a significant effect on the increase in lap shear strength. Moreover, the failure mainly occurs at the overmolded interface at lower injection temperature and preheating temperature. As the injection temperature and preheating temperature increase, the failure occurs at both the co-cured interface and overmolded interface. It means that the bonding strength of the co-cured interface is greater than that of the overmolded interface.In the lap shear test, the maximum lap shear strength reaches 24.4 ± 4.2 MPa, and the co-cured interface does not completely fail. The strength is greater than the 10.7 ± 0.3 MPa of the co-cured interface before injection overmolding process. It means that the bonding strength of the co-cured interface has been enhanced after the injection overmolding process.Molecular dynamics simulation was carried out to investigate the mechanism of the enhancement of the co-cured interface. The simulation result shows that high temperature and pressure during the injection overmolding process can only weakly increase the bonding strength of the co-cured interface Moreover, the interfacial chemical reaction is the main reason for the enhancement of the co-cured interface. Specifically, the chemical reactions at the interface can significantly increase the interfacial bonding energy and maximum tensile strength, and the enhancement effect increases with the increase in the number of reaction atoms.

## Figures and Tables

**Figure 1 polymers-15-02879-f001:**
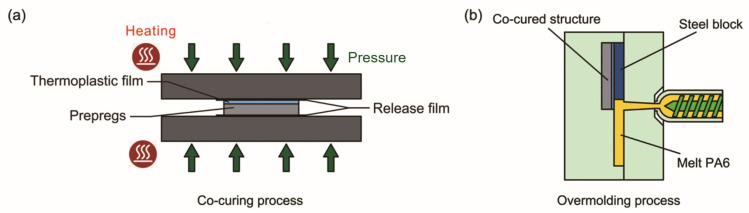
Schematic diagram of: (**a**) co-curing process and (**b**) injection overmolding process [18].

**Figure 2 polymers-15-02879-f002:**
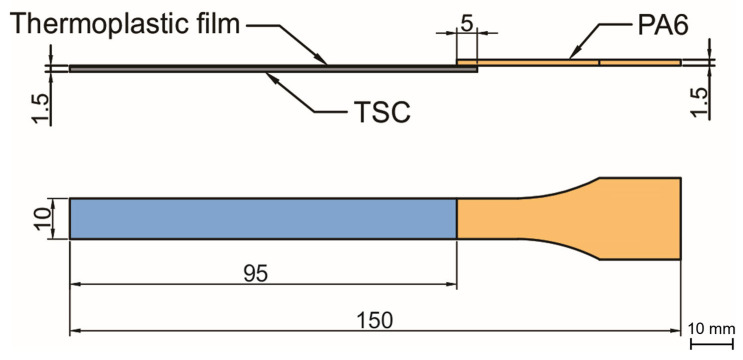
Schematic depiction of overmolded hybrid thermoset–thermoplastic sample [18].

**Figure 3 polymers-15-02879-f003:**
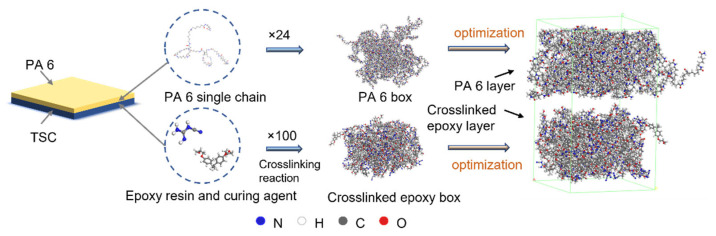
Conformations of atomistic model of the interface between crosslinked epoxy resin and PA 6.

**Figure 4 polymers-15-02879-f004:**
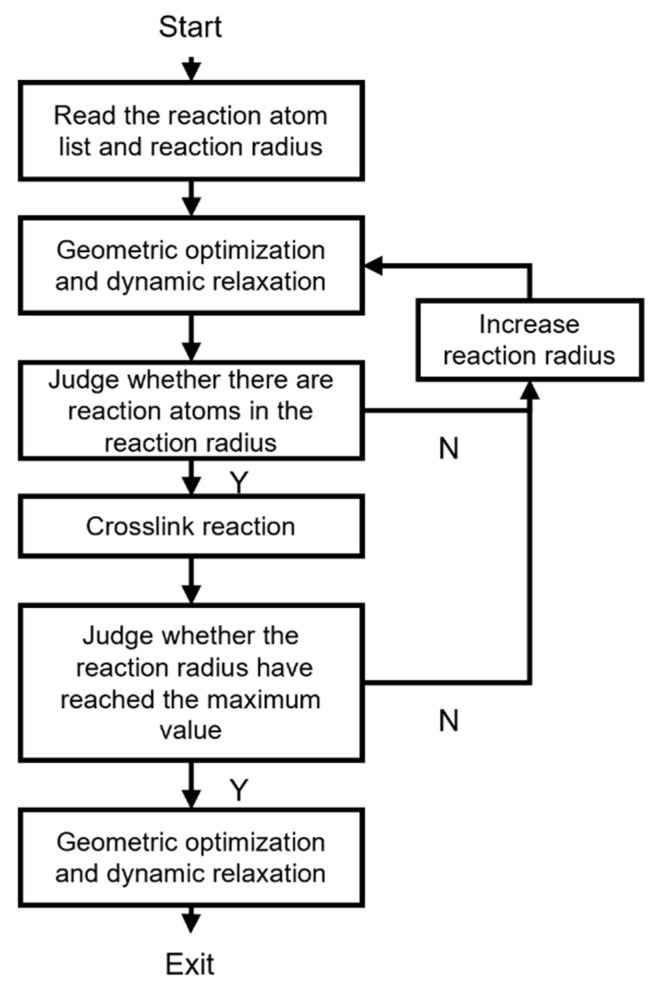
DGEBA and DICY crosslink reaction flowchart.

**Figure 5 polymers-15-02879-f005:**
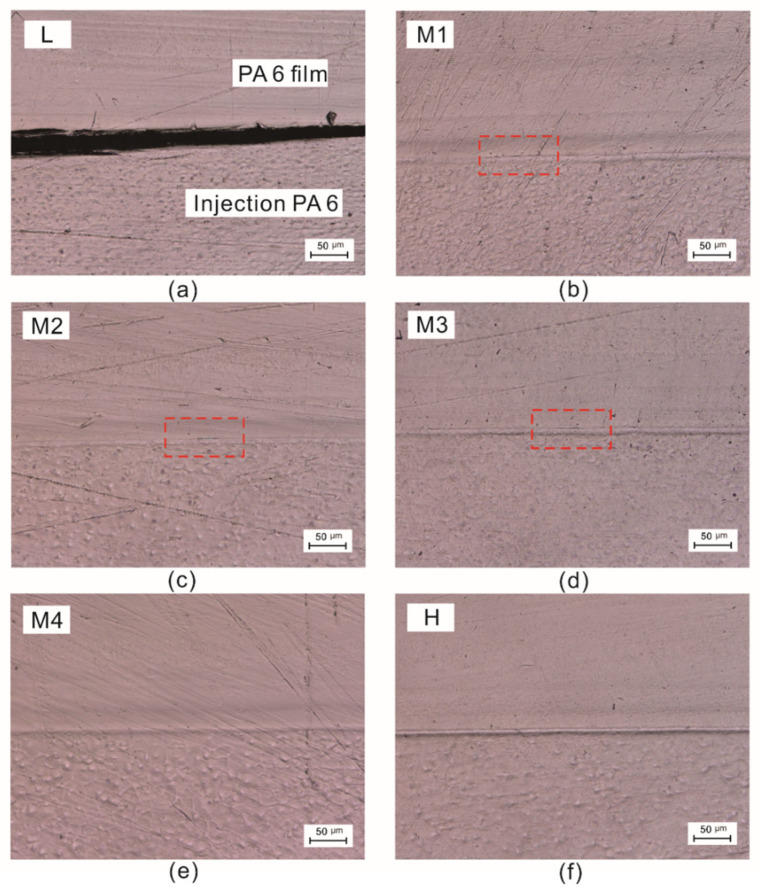
The cross-section micrographs of the overmolded interfaces of (**a**) L, (**b**) M1, (**c**) M2, (**d**) M3, (**e**) M4, (**f**) H samples.

**Figure 6 polymers-15-02879-f006:**
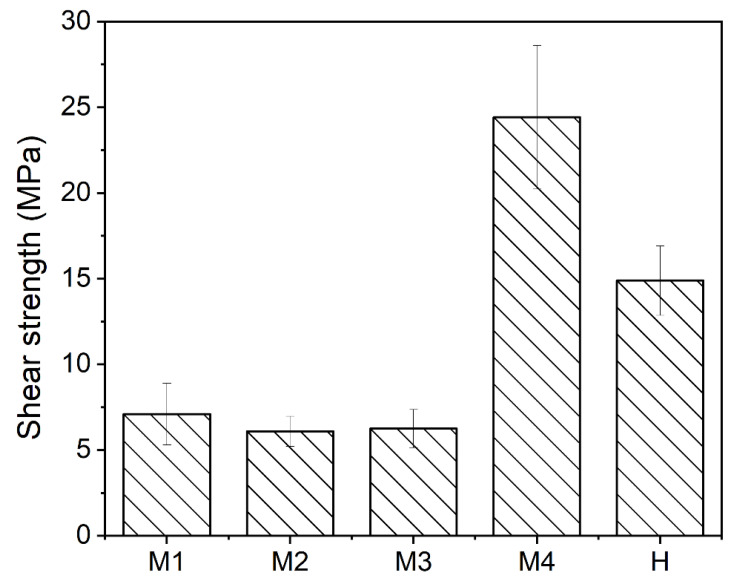
The lap shear strength of TSC–PA 6 structure fabricated with different process parameters.

**Figure 7 polymers-15-02879-f007:**
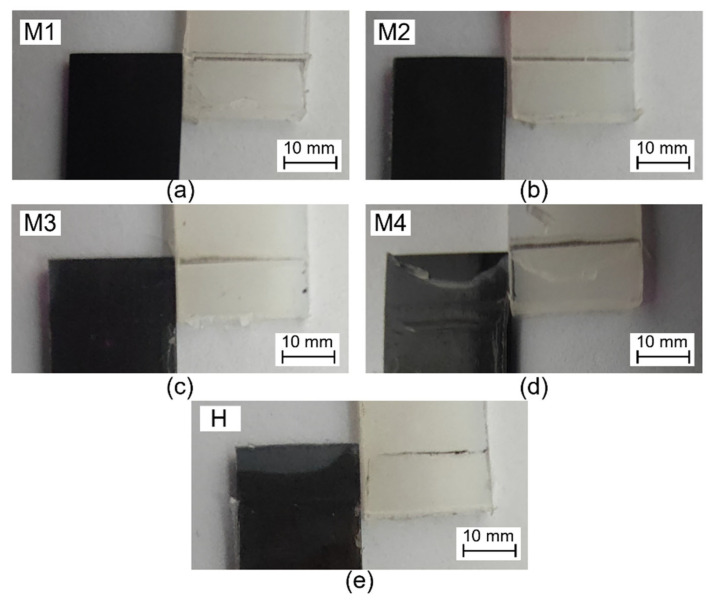
The fracture surface pattern of (**a**) M1, (**b**) M2, (**c**) M3, (**d**) M4, (**e**) H samples.

**Figure 8 polymers-15-02879-f008:**
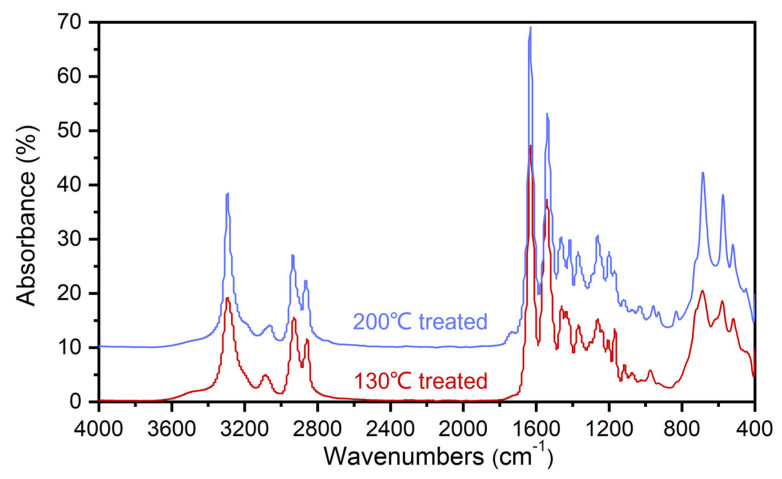
FTIR spectra of the treated samples surface.

**Figure 9 polymers-15-02879-f009:**
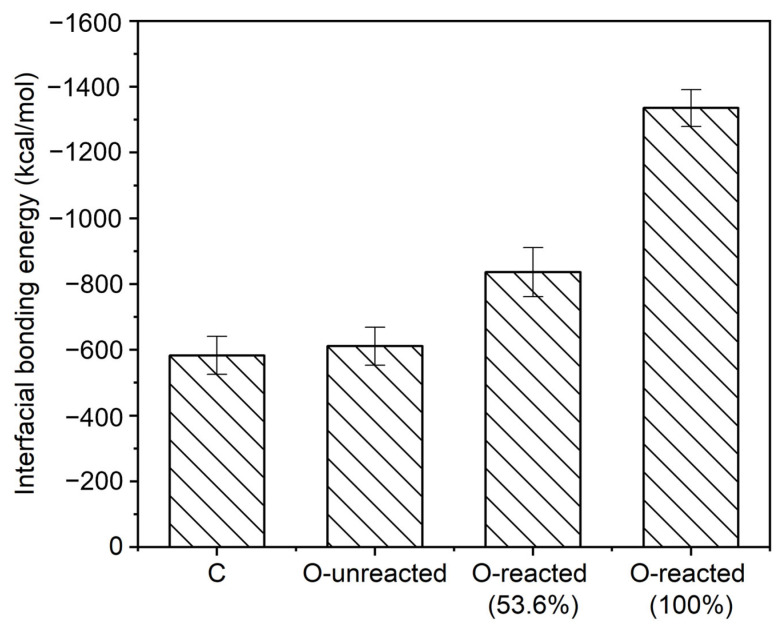
The interfacial bonding energy between PA 6 and crosslinked epoxy resin.

**Figure 10 polymers-15-02879-f010:**
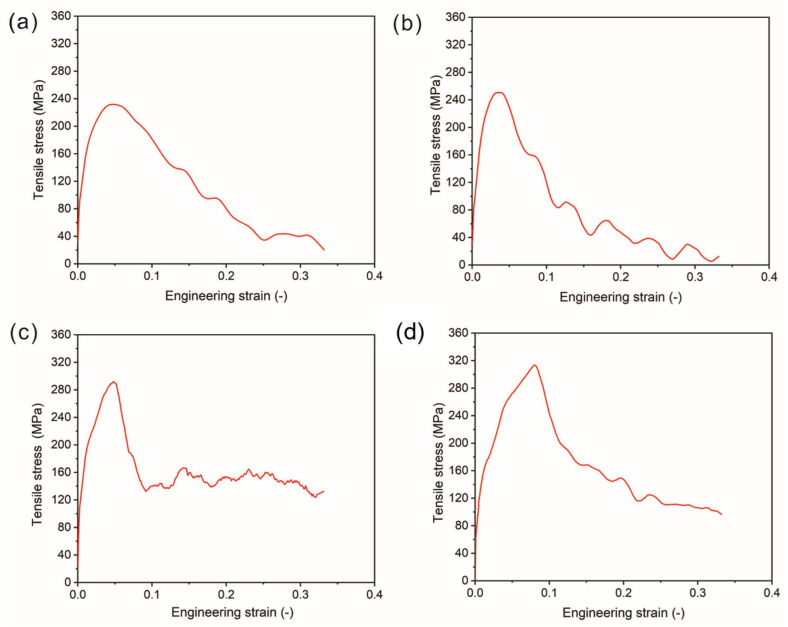
The stress vs. strain curves for (**a**) C, (**b**) O-unreacted, (**c**) O-reacted (53.6%) and (**d**) O-reacted (100%) interfaces.

**Figure 11 polymers-15-02879-f011:**
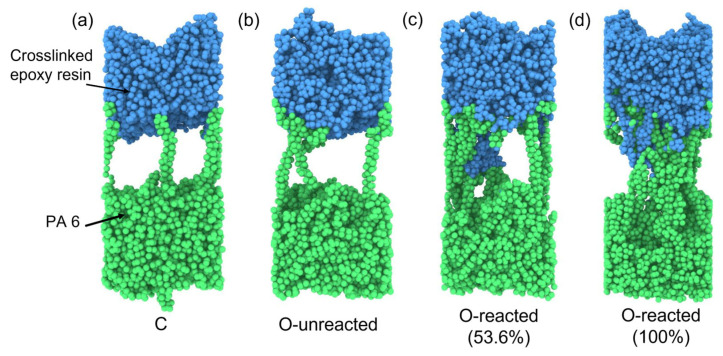
Snapshots of the failure modes of (**a**) C, (**b**) O-unreacted, (**c**) O-reacted (53.6%), (**d**) O-reacted (100%) interfaces.

**Table 1 polymers-15-02879-t001:** The overmolding process parameter used in this study.

Name	Injection Temperature (°C)	Injection Speed (mm/s)	Preheating Temperature (°C)	Holding Pressure (MPa)	Cooling Time (s)
H	280	60	80	20	40
M1	260	30	80		
M2	260	60	80		
M3	260	90	80		
M4	260	60	150		
L	240	60	80		

**Table 2 polymers-15-02879-t002:** Forming conditions for four groups of interface models.

Name	Co-Curing Process	Overmolding Process	Reaction 1 Reaction Amount	Reaction 2 Reaction Amount
C	400 K 0.6 MPa	/	/	/
O-unreacted		500 K 20 MPa	/	/
O-reacted (53.6%)			1	15
O-reacted (100%)			1	28

## Data Availability

The data presented in this study are available on request from the corresponding author.

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
