# Peer review of "Effect of Injection Overmolding Parameters on the Interface Bonding Strength of Hybrid Thermoset–Thermoplastic Composites"

_polymers, 2023, doi:10.3390/polym15132879_

Round 1

Reviewer 1 Report

Dear Authors,

This is an interesting and relevant study that could be published after major revisions.

The results are innovative and the methodology is sound.

The conclusions are supported by the results.

Before the paper can be considered for publication, Authors should address the following comments:

- The English language must be revised by a native English editor typesetter before publication

- In §3.2. Single lap shear strength of TSC-PA 6 structure

Please report the lap shear strength results of H, M1, M2, M3 and 289 M4 samples as follows: 7.1 ± 1.8 MPa, 6.1 ± 0.9 MPa, 6.25 ± 1.1 MPa, 24.4 ± 4.2 MPa and 14.9 ± 2.0 MPa, respectively.

Please also report the maximum lap shear strength results of co-curing interface between PA 6 film and TSC as follows: 10.7 ± 0.3 MPa, 24.4 ± 4.2 MPa.

- Add scale bars in Figure 7

- Absorbance values on the ve rtical axis in Figure 8 missing

- In section 3.4 the interfacial bonding energy results of C, O-unreacted, O-reacted (53.6%) and O-reacted (100%) interface models should be indicated as follows: -583 ± 58, -611 ± 58, -836 ± 74 and -1336 ± 56 349 Kcal/mol, respectively.

- In section 3.4.2.1 please correct the sentence 'It can be concluded by comparing the result of the maximum tensile stress that, the maximum tensile strength of the O-unreacted model increased from 241.22 MPa for C model to 250.69 MPa. It indicates that the interfacial bonding strength has been improved under high temperature and pressure, but the increase is relatively limited. In contrast, the maximum tensile strength of the O-reacted (53.6%) and O-reacted (100%) models increase to 291.92 MPa and 317.35 MPa. It indicates that chemical bonding has a significant effect on improving interfacial bonding strength, and the reinforcement effect increases with the increase in the amounts of chemical reactions.' as follows: 'It can be concluded by comparing the result of the maximum tensile stress that, the maximum tensile strength of the O-unreacted model increased from 241 MPa for C model to 251 MPa. It indicates that the interfacial bonding strength has been improved under high temperature and pressure, but the increase is relatively limited. In contrast, the maximum tensile strength of the O-reacted (53.6%) and O-reacted (100%) models increase to 291.92 MPa and 317.35 MPa. It indicates that chemical bonding has a significant effect on improving interfacial bonding strength, and the reinforcement effect increases with the increase in the amounts of chemical reactions.' - In addition, please complete the two results with their corresponding standard deviation.

- What is the repeatability (e.g., standard deviation) of the plots reported in Figure 10.

- Sextion 4. Conclusions should be first expanded and then written more clearly, for example by means of a bullet points list.

- In section 4 the sentence 'Besides, the maximum lap shear strength of TSC-PA 6 structure is 24.43 ± 4.19 MPa, which is far more than 10.68 ± 0.32 MPa of the co-cured structure. It means that the bonding strength of co-cured interface has been enhanced after injection overmolding process.' should be changes as follows: 'Besides, the maximum lap shear strength of TSC-PA 6 structure is 24.4 ± 4.2 MPa, which is far more than 10.68 ± 0.32 MPa of the co-cured structure. It means that the bonding strength of co-cured interface has been enhanced after injection overmolding process.'

The English language must be revised by a native English editor typesetter before publication

Author Response

Point 1: The English language must be revised by a native English editor typesetter before publication.

Response 1: Thank you for your careful review. Action taken as suggested.

Point 2: In §3.2. Single lap shear strength of TSC-PA 6 structure Please report the lap shear strength results of H, M1, M2, M3 and 289 M4 samples as follows: 7.1 ± 1.8 MPa, 6.1 ± 0.9 MPa, 6.25 ± 1.1 MPa, 24.4 ± 4.2 MPa and 14.9 ± 2.0 MPa, respectively.

 Please also report the maximum lap shear strength results of co-curing interface between PA 6 film and TSC as follows: 10.7 ± 0.3 MPa, 24.4 ± 4.2 MPa.

Response 2: Thanks for your careful reading. The sentences have been modified, please find it in the revised manuscript.

Point 3: Add scale bars in Figure 7.

Response 3: Thank you for your reminder. Action taken as suggested.

Point 4: Absorbance values on the vertical axis in Figure 8 missing.

Response 4: Thanks for your reminder. The absorbance values on the vertical axis in Figure 8 has been added.

Point 5: In section 3.4 the interfacial bonding energy results of C, O-unreacted, O-reacted (53.6%) and O-reacted (100%) interface models should be indicated as follows: -583 ± 58, -611 ± 58, -836 ± 74 and -1336 ± 56 349 Kcal/mol, respectively.

Response 5: Thank you for your careful reading. The interfacial bonding energy results have been modified, please find it in the revised manuscript.

Point 6: In section 3.4.2.1 please correct the sentence 'It can be concluded by comparing the result of the maximum tensile stress that, the maximum tensile strength of the O-unreacted model increased from 241.22 MPa for C model to 250.69 MPa. It indicates that the interfacial bonding strength has been improved under high temperature and pressure, but the increase is relatively limited. In contrast, the maximum tensile strength of the O-reacted (53.6%) and O-reacted (100%) models increase to 291.92 MPa and 317.35 MPa. It indicates that chemical bonding has a significant effect on improving interfacial bonding strength, and the reinforcement effect increases with the increase in the amounts of chemical reactions.' as follows: 'It can be concluded by comparing the result of the maximum tensile stress that, the maximum tensile strength of the O-unreacted model increased from 241 MPa for C model to 251 MPa. It indicates that the interfacial bonding strength has been improved under high temperature and pressure, but the increase is relatively limited. In contrast, the maximum tensile strength of the O-reacted (53.6%) and O-reacted (100%) models increase to 291.92 MPa and 317.35 MPa. It indicates that chemical bonding has a significant effect on improving interfacial bonding strength, and the reinforcement effect increases with the increase in the amounts of chemical reactions.' - In addition, please complete the two results with their corresponding standard deviation.

Response 6: Thank you for your attention to detail in reading. The sentences have been rewritten. Please find it in the revised manuscript. While, the results shown in Figure 10 were obtained from molecular dynamic simulation. When the model is fixed, the simulation results of the model are same with the same boundary condition regardless of calculation times. Therefore, there is no the standard deviation in Figure 10. Same actions were also taken in literatures about molecular dynamics simulation [1-6].

Point 7: What is the repeatability (e.g., standard deviation) of the plots reported in Figure 10.

Response 7: The results of Figure 10 are obtained from molecular dynamics simulation. For molecular dynamic simulation, the tensile stress results are the same when the model is fixed. Therefore, there is no standard deviation in Figure 10.

Point 8: Section 4. Conclusions should be first expanded and then written more clearly, for example by means of a bullet points list.

Response 8: Thank you for your suggestion. The conclusions have been expanded and rewritten.

Point 9:  In section 4 the sentence 'Besides, the maximum lap shear strength of TSC-PA 6 structure is 24.43 ± 4.19 MPa, which is far more than 10.68 ± 0.32 MPa of the co-cured structure. It means that the bonding strength of co-cured interface has been enhanced after injection overmolding process.' should be changes as follows: 'Besides, the maximum lap shear strength of TSC-PA 6 structure is 24.4 ± 4.2 MPa, which is far more than 10.68 ± 0.32 MPa of the co-cured structure. It means that the bonding strength of co-cured interface has been enhanced after injection overmolding process.'

Response 9: Thanks for your careful reading. The sentences have been rewritten. Please find it in the revised manuscript.

References

  1. Izadi, R., et al., Thermomechanical characteristics of green nanofibers made from polylactic acid: An insight into tensile behavior via molecular dynamics simulation. Mechanics of materials, 2023. 181: p. 104640.
  2. Yang, S. and J. Qu, Coarse-grained molecular dynamics simulations of the tensile behavior of a thermosetting polymer. Physical review. E, Statistical, nonlinear, and soft matter physics, 2014. 90(1): p. 012601-012601.
  3. Liu, H., et al., Insights into the tensile behavior of polymer nanofibers with hierarchically twisted chains. Computational materials science, 2021. 194: p. 110463.
  4. Zhou, M., et al., Molecular Dynamics Simulation on the Effect of Bonding Pressure on Thermal Bonding of Polymer Microfluidic Chip. Polymers, 2019. 11(3): p. 557.
  5. Jiang, B., et al., Molecular Dynamics Simulation on the Interfacial Behavior of Over-Molded Hybrid Fiber Reinforced Thermoplastic Composites. Polymers, 2020. 12(6): p. 1270.
  6. Zhang, M., et al., The effect of self-resistance electric heating on the interfacial behavior of injection molded carbon fiber/polypropylene composites through molecular dynamics analysis. Polymer (Guilford), 2020. 207: p. 122915.

Reviewer 2 Report

Comment 1: The authors are discussing the interfacial bonding strength between epoxy and overmolded polyamide. They widely use the term “interface” in their study. However, as per my understanding in their study, there are 2 interfaces: one is the so-called “co-cured interface” formed between epoxy and PA film, and another one is between PA film and overmolded PA. The formation of the first interface (between epoxy and PA film) is mostly governed by the chemical reaction, while the formation of the second interface (PA film and overmolded PA) is mostly governed by the molecular interdiffusion process. So, it is not clear (especially in section 2.4.2.1) what kind of interface they are talking about. Please, first clarify what you mean under the "interface" in this study. And then, please clearly specify, which kind of interface you are talking about in each part of the article.

Comment 2: The authors mentioned “degree of healing theory” in their study. In fact, they are talking about the theory of reptation developed by de Gennes. So, it is recommended to correct the name of the theory they refer to. There are also several recent articles dedicated to the development of the reptation theory, and it is worth adding them to the literature overview in order to give a better state-of-the-art to the readers:

https://doi.org/10.1016/j.compositesb.2016.04.064

https://doi.org/10.1080/10426914.2021.1948052

https://doi.org/10.3389/fmats.2020.00027

https://doi.org/10.1002/app.50294

Comment 3: the manuscript contains grammatical mistakes. For example, lines 45-47: “In contrast, placing TSC into the injection mold and directly bonding the TSC to thermoplastic material through injection overmolding show greater potential in reduce reduction the cycle time… “

Line 72: co-curd interface, etc.

The Authors are recommended to do professional English proofreading.

the manuscript contains grammatical mistakes. For example, lines 45-47: “In contrast, placing TSC into the injection mold and directly bonding the TSC to thermoplastic material through injection overmolding show greater potential in reduce reduction the cycle time… “

Line 72: co-curd interface, etc.

The Authors are recommended to do professional English proofreading

Author Response

Point 1: The authors are discussing the interfacial bonding strength between epoxy and overmolded polyamide. They widely use the term “interface” in their study. However, as per my understanding in their study, there are 2 interfaces: one is the so-called “co-cured interface” formed between epoxy and PA film, and another one is between PA film and overmolded PA. The formation of the first interface (between epoxy and PA film) is mostly governed by the chemical reaction, while the formation of the second interface (PA film and overmolded PA) is mostly governed by the molecular interdiffusion process. So, it is not clear (especially in section 2.4.2.1) what kind of interface they are talking about. Please, first clarify what you mean under the "interface" in this study. And then, please clearly specify, which kind of interface you are talking about in each part of the article.

Response 1: It is indeed that there are two interfaces in this study, namely co-cured interface and overmolded interface. Specifically, the co-cured interface refers to the interface between thermoset composite (TSC) and PA 6 film, while the overmolding interface refers to the interface between PA 6 film and overmolded PA 6. Moreover, the co-cured interface is simplified as the interface between resin and PA 6 film in molecular dynamic simulation. The interface has been clarified in the revised manuscript as suggested. Please find in the revised manuscript.

Point 2: The authors mentioned “degree of healing theory” in their study. In fact, they are talking about the theory of reptation developed by de Gennes. So, it is recommended to correct the name of the theory they refer to. There are also several recent articles dedicated to the development of the reptation theory, and it is worth adding them to the literature overview in order to give a better state-of-the-art to the readers:

https://doi.org/10.1016/j.compositesb.2016.04.064

https://doi.org/10.1080/10426914.2021.1948052

https://doi.org/10.3389/fmats.2020.00027

https://doi.org/10.1002/app.50294

Response 2: Action taken as suggested.

Point 3:  the manuscript contains grammatical mistakes. For example, lines 45-47: “In contrast, placing TSC into the injection mold and directly bonding the TSC to thermoplastic material through injection overmolding show greater potential in reduce reduction the cycle time… “

Line 72: co-curd interface, etc.

The Authors are recommended to do professional English proofreading.

Response 3: Thank you for your attention to detail in reading. Action taken as suggested.
